# Characteristics of Polyurethane/Waste Rubber Powder Composite Modifier and Its Effect on the Performance of Asphalt Mixture

**Bo Gao, Yuechao Zhao * and Zenggang Zhao**

State Key Laboratory of Silicate Materials for Architectures, Wuhan University of Technology, Wuhan 430070, China; 18627808373@163.com (B.G.); zhaozenggang@whut.edu.cn (Z.Z.)
* Correspondence: zhaoyc@whut.edu.cn

**Abstract:** To solve the problems of storage stability and the volatile organic compound emission of waste-rubber-powder-modified bitumen, the strategy of preparing composite modifiers using waterborne polyurethane coating for waste rubber powder was proposed in an early-stage work. However, the effect of polyurethane/waste rubber powder composite modifier on the performance of asphalt mixture is unclear, which limits the further popularization of this technology. Therefore, this work mainly investigates the characteristics of composite modifiers and their influence on asphalt mixture. The results indicate that the optimum ratio of polyurethane to waste rubber powder is about 1:1, which can give the composite modifier sufficient mechanical properties and acceptable economic benefits. The scanning electron microscopy images also confirm that polyurethane can densely encapsulate waste rubber powder. The physical properties experiment of bitumen shows that composite modifiers can increase the softening point and viscosity of bitumen but reduce the ductility and penetration of bitumen. Moreover, it was also found that composite modifiers can significantly improve asphalt mixtures' resistance to permanent deformation and moisture damage. This can be attributed to the increase in the consistency of the asphalt binder due to the composite modifier. However, the anti-cracking properties of asphalt mixtures will be destroyed if the content of the composite modifier is too high. Therefore, it is necessary to balance the high and low temperature properties of asphalt mixtures when determining the dosage of composite modifiers in practical engineering. The results of this paper can provide a reference for the green application of waste-rubber-powder-modified bitumen.

**Keywords:** polyurethane; waste rubber powder; asphalt; composite modification

## 1. Introduction

Since the beginning of the new century, the good economic form and broad market environment have promoted the rapid development of China's automobile industry [1]. The number of motor vehicles in China reached 402 million at the end of March 2022 according to the National Bureau of Statistics [2]. The rapid growth of the automobile field has brought convenience to society but has also put forward more challenges [3]. The renewal of automobiles has produced a large number of waste tires, which poses a great threat to the ecological environment, which is known to the population as "black pollution" [4]. Waste tires are highly resistant to weathering and will not naturally disappear for decades. The open piling of end-of-life tires not only takes up a lot of land resources, but also breeds mosquitoes and infectious diseases easily, as well as causes fires [5]. Meanwhile, modern traffic with high traffic volumes and heavy loads also places higher performance requirements on roads, especially asphalt pavement [6]. A best-of-both-worlds approach is to grind waste tires into powder for asphalt modification, which not only helps to relieve the pressure of waste tire disposal, but also improves the durability of traditional asphalt pavement to a certain extent to meet the challenges of modern traffic [7,8].

At present, whether from the perspective of waste disposal or simply to improve the performance of bitumen, many scholars have undertaken work on the modification of bitumen using waste rubber powder [9,10]. It was found that waste rubber powder has many advantages as an asphalt modifier, such as improving the durability, fatigue resistance, and economic benefits of asphalt pavement [11–13]. Li et al. pointed out that the aging resistance of waste-tire-rubber-powder-modified asphalt is also excellent [14]. The excellent flexibility of rubber also contributes to enhancing the ability of asphalt pavement to resist cracking in cold regions [15,16]. Another study also indicates that asphalt mixtures modified with waste rubber powder have excellent noise reduction functions, which can improve driving comfort [17,18]. In addition, road construction has the characteristics of high resource occupation [19], which can consume the accumulated waste tires on a large scale. In conclusion, road engineering is a potential field for recycling waste rubber.

Despite the effectiveness of waste rubber powder to improve the performance of asphalt pavements, there are still two more difficult problems with rubber-powder-modified asphalt [20–22]. Asphalt, as an organic material, releases a lot of chemical fumes during the construction process, and the introduction of waste rubber powder further worsens the situation [23]. On the other hand, the poor compatibility of waste rubber powder with bitumen is a consensus in this field [24–26]. The separation phenomena are more obvious when the dosage of waste rubber powder in asphalt is higher [27]. To solve the above two problems simultaneously, waterborne polyurethane was used to encapsulate the waste rubber powder in our preliminary work [1]. Polyurethane is a designable material that is adapted to improve compatibility with asphalt by changing its formulation [28]. In addition, most of the fumes released by the rubber powder due to high-temperature construction are shielded within the polyurethane film. This innovative method is very effective in improving the compatibility of waste rubber powder with bitumen and inhibiting the emission of flue gas.

However, the performance of this new composite modifier in asphalt mixtures is not yet clear, which hinders its large-scale industrial application. Therefore, the main objectives of this study are to explore the characteristics of polyurethane/waste rubber powder composite modifiers and their effect on the performance of asphalt mixtures. For this purpose, the optimal mixing ratio of waterborne polyurethane and waste rubber powder was first determined via film stretching experiments. The microscopic morphology of polyurethane/waste rubber powder composite modifiers was also observed via scanning electron microscopy. Additionally, the influence of polyurethane/waste rubber powder composite modifiers on the engineering performance of asphalt mixtures was also evaluated, including deformation resistance, moisture susceptibility, cracking resistance, etc. The conclusions of the study can provide a reference for the resource utilization of waste tire rubber powder and the green development of road engineering. The research roadmap for this work is presented in Figure 1.

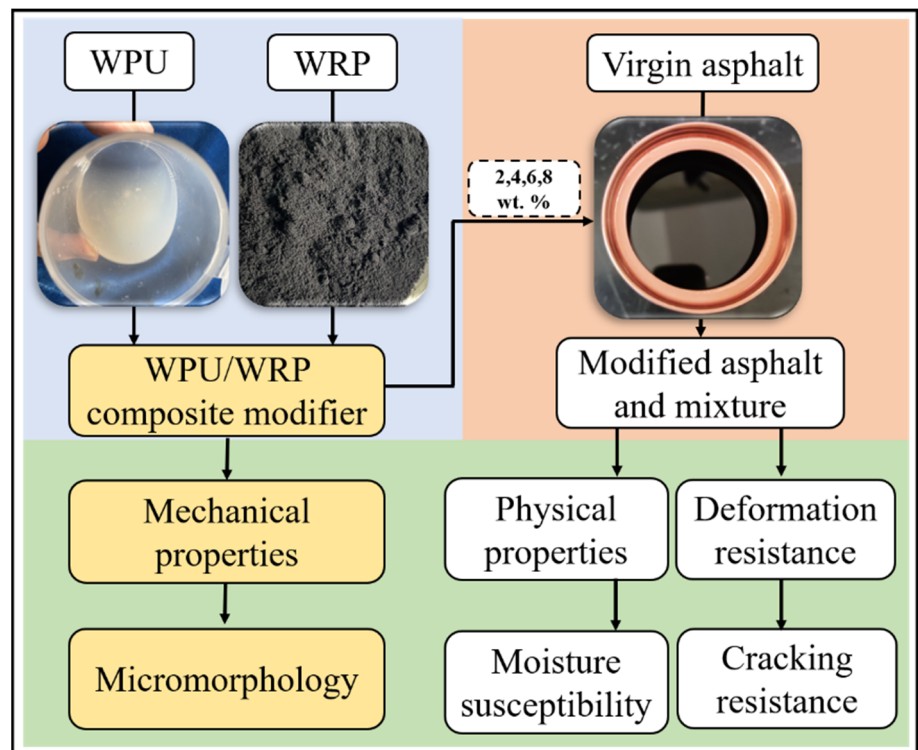

**Figure 1.** The research roadmap for this work.

## 2. Materials and Methods

### 2.1. Raw Materials

Petroleum asphalt with a penetration grade of 70# of the Donghai brand was selected as the virgin asphalt in this work. Its basic properties including softening point, ductility, penetration, and density are illustrated in Table 1. The aggregate applied to prepare asphalt mixtures was basalt sourced from Jingshan City, Hubei Province in China. The physical parameters of the aggregate are given in Table 2. The waste rubber powder with a mesh size of 150 was obtained from a waste tire disposal plant. Sulfonic acid type waterborne polyurethane was synthesized in the laboratory, which has a high solid content, high bond strength, and obvious microphase separation between soft segments and hard segments. Its functional group information is shown in Figure 2. The detailed preparation method can be found in our previous works [1].

**Table 1.** The basic properties of virgin asphalt.

| Properties | Test Values | Test Method (JTG E20-2011 [29]) |
|---|---|---|
| Penetration | 67 (0.1 mm) | T 0604 |
| Softening point | 49.7 (°C) | T 0606 |
| Ductility | >100 (cm) | T 0605 |
| Density | 1.022 (g/cm$^3$) | T 0603 |

**Table 2.** The basic physical properties of aggregate.

| Items | Grain Size (mm) | Results | Requirements |
|---|---|---|---|
| Apparent specific gravity | 9.5~16 | 3.032 | ≥2.6 |
| | 4.75~9.5 | 2.998 | |
| | 2.36~4.75 | 3.009 | |
| | <2.36 | 2.987 | |

**Table 2.** *Cont.*

| Items | Grain Size (mm) | Results | Requirements |
|---|---|---|---|
| Water absorption (%) | 9.5~16 | 0.53 | ≤3 |
| | 4.75~9.5 | 0.62 | |
| | 2.36~4.75 | 0.98 | |
| Los Angeles abrasion (%) | | 14.2 | ≤28 |
| Crushed value (%) | | 15.1 | ≤26 |

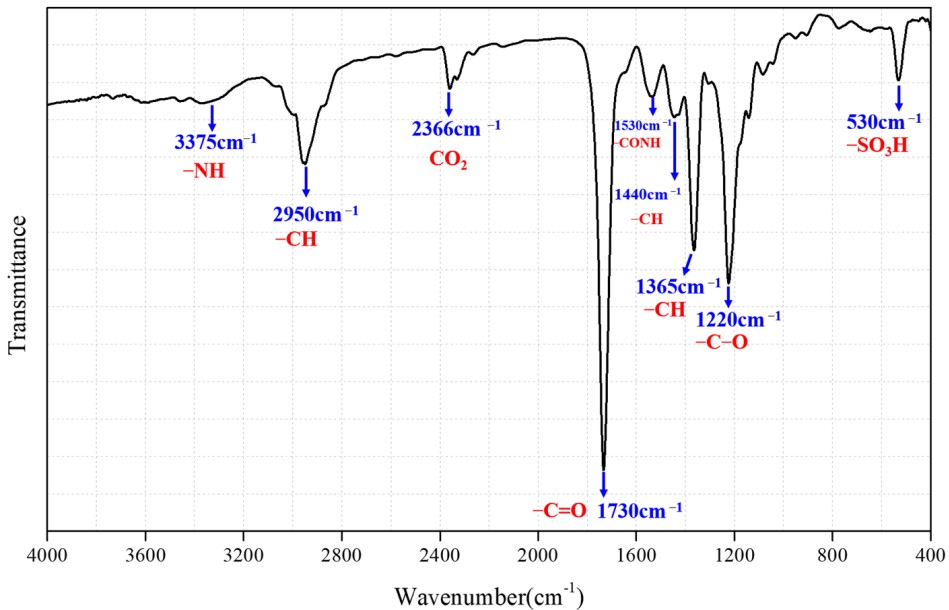

**Figure 2.** The infrared spectrum of waterborne polyurethane.

### 2.2. Samples Preparation

#### 2.2.1. Polyurethane/Waste Rubber Powder Composite Modifier

Waterborne polyurethane is an emulsion in which polyurethane molecular chains are dispersed in water. Therefore, waste rubber powder can be directly mixed into this emulsion system via mechanical agitation. With the evaporation of water, the emulsion will gradually solidify to form a composite material of polyurethane and waste rubber powder. Firstly, a certain amount of waste rubber powder was added to the waterborne polyurethane solution. Afterward, the above mixture was stirred thoroughly with a glass rod so that each piece of waste rubber powder was covered with polyurethane. Finally, the homogeneously mixed polyurethane/waste glue powder solution was extruded through a syringe to dry in the form of small particles. Four kinds of composite modifiers were prepared by adjusting the ratio of polyurethane to waste rubber powder. Specifically, the ratios of polyurethane to rubber powder were 1:0; 3:2; 1:1; and 2:3, named PR10, PR32, PR11, and PR23, respectively.

#### 2.2.2. Modified Asphalt Binder

Five kinds of modified binders with different composite modifier contents were prepared using the melt blending method. Five groups of modifier content were set including 0.0 wt%, 2.0 wt%, 4.0 wt%, 6.0 wt%, and 8.0 wt% of virgin asphalt mass. The shear temperature, time, and rate of the modifier and virgin asphalt were set at 170 °C, 1 h, and 6000 rpm, respectively.

#### 2.2.3. Modified Asphalt Mixture

The influence of composite modifiers on the behavior of asphalt mixtures was verified via the continuous dense gradation of AC-13, which is commonly applied for the surface

layer of high-grade highways. The optimal ratio of bitumen to stone was designed to be 4.50% based on the Marshall design method. The gradation curves are shown in Figure 3.

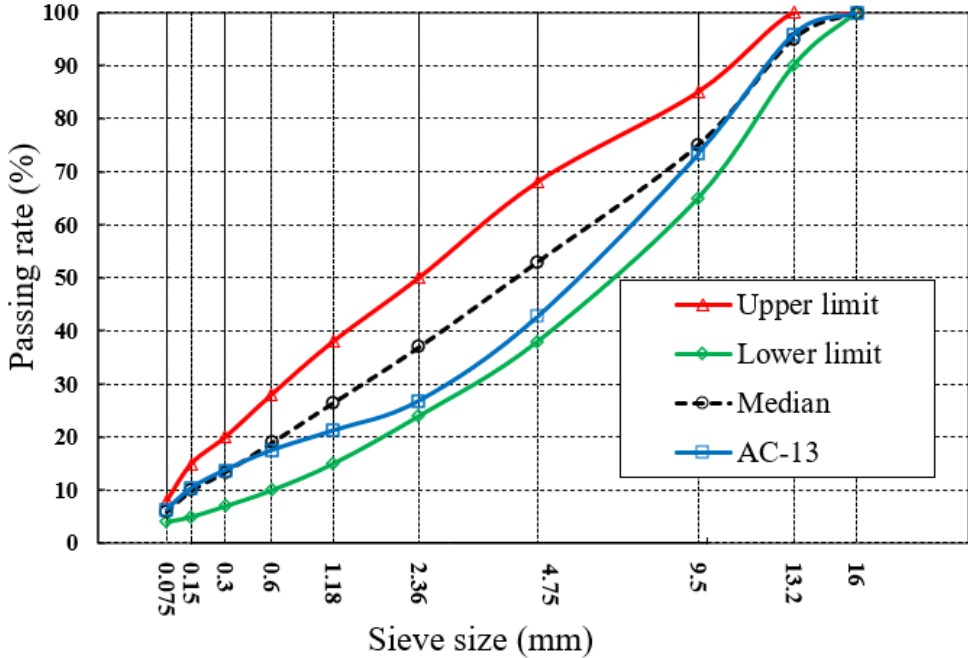

**Figure 3.** Gradation curves of AC-13.

*2.3. Measurement and Characterization*

2.3.1. Tensile Strength Test of Composite Modifier

The tensile property of the composite modifier was measured through a universal test machine (Instron 5967) at a loading speed of 50 mm/min according to the Chinese standard of GB/T 528-1998 [30]. The test sample was a type I dumbbell shape with a cross-section size of 33 mm × 6 mm × 2 mm [31], as shown in Figure 4. Three effective parallel experiments were performed on each sample.

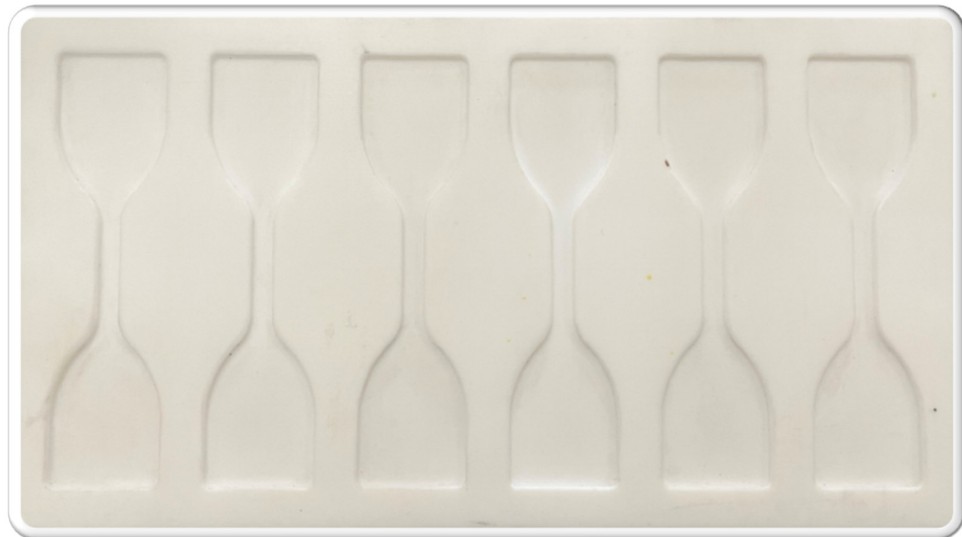

**Figure 4.** The mold for tensile property experiment.

2.3.2. Scanning Electron Microscope Test of Modified Modifier

The microscopic morphology of waste rubber powder before and after being coated with waterborne polyurethane was observed using a scanning electron microscope (Quanta

FEG). The scanning was carried out at a voltage of 15 KV with gold powder as the conductive medium.

### 2.3.3. Basic Properties Test of Modified Asphalt Binders

The basic physical properties of the bitumen modified with composite modifier, such as softening point, penetration (25 °C), ductility (15 °C), and viscosity (135 °C), were carried out regarding the Chinese standard JTG E20-2011.

### 2.3.4. Deformation Resistance of Modified Asphalt Mixture

The effects of composite modifiers on the permanent deformation resistance of asphalt pavements were determined through wheel tracking experiments. Samples with dimensions of 300 mm × 300 mm × 50 mm were formed and kept in an environment of 60 °C for at least 5.0 h. With the experiment run, a rubber wheel acted on the surface of the specimen with a pressure of 0.7 MPa at a frequency of 42 times/min [29]. The deformation resistance of the asphalt mixture can be expressed by its dynamic stability, which is the number of wheel rolls required for every 1 mm deformation of the asphalt mixture specimen during the experiment. The higher the value of the dynamic stability, the greater the deformation resistance of the asphalt mixture in high-temperature environments.

### 2.3.5. Cracking Resistance of Modified Asphalt Mixture

Three-point bending experiments were conducted to evaluate the influence of composite modifiers on the anti-cracking properties of asphalt pavements. Samples with dimensions of 250 mm × 30 mm × 35 mm were kept in an environment of −10 °C for at least 4 h. The indenter of the universal testing machine acted vertically on the center of the samples at a loading rate of 50 mm/min until it fractured [29]. The anti-cracking properties of asphalt mixtures can be characterized by the ultimate flexural strain. Generally, the lower the ultimate flexural strain is, the worse the toughness of the asphalt mixture is.

### 2.3.6. Moisture Susceptibility of Modified Asphalt Mixture

In this work, the influence of composite modifiers on the moisture susceptibility of asphalt pavements was determined via residual Marshall stability (*RMS*) experiments and the indirect tensile strength ratio (*ITSR*). It should be noted that the Marshall specimen used for stability testing requires 75 compaction times during molding, while the Marshall specimen used for indirect tensile strength testing requires 50 compaction times during molding.

The *RMS* and *ITSR* are calculated as follows [29]:

$$RMS = \frac{MS_1}{MS_0} \times 100\%. \tag{1}$$

$$ITSR = \frac{TS_1}{TS_0} \times 100\% \tag{2}$$

where $MS_1$ and $MS_0$ are the stability after soaking the Marshall specimens in water for 35 min and 48 h, respectively, and $TS_1$ and $TS_0$ are the tensile strengths of unfreeze–thawed specimens and freeze–thawed specimens, respectively.

## 3. Results and Discussion

### 3.1. Properties of Polyurethane/Waste Rubber Powder Composite Modifier

#### 3.1.1. Tensile Strength

The main advantage of the composite modifier is that it prevents direct contact between bitumen and waste rubber powder, which reduces the number of fumes emitted from the waste rubber powder to the outside world and improves the compatibility of the modifier with the asphalt. Therefore, it is necessary to ensure that the structure of the composite modifier is not destroyed in the process of preparing the modified asphalt. That is, polyurethane can still be tightly wrapped around the waste rubber powder after high-speed mechanical mixing. The strength of the composite modifier is mainly determined by

the proportions of polyurethane and waste rubber powder. To determine the best formula for composite modifiers, the strength of different composite modifiers was measured using the tensile test. The curves are presented in Figure 5.

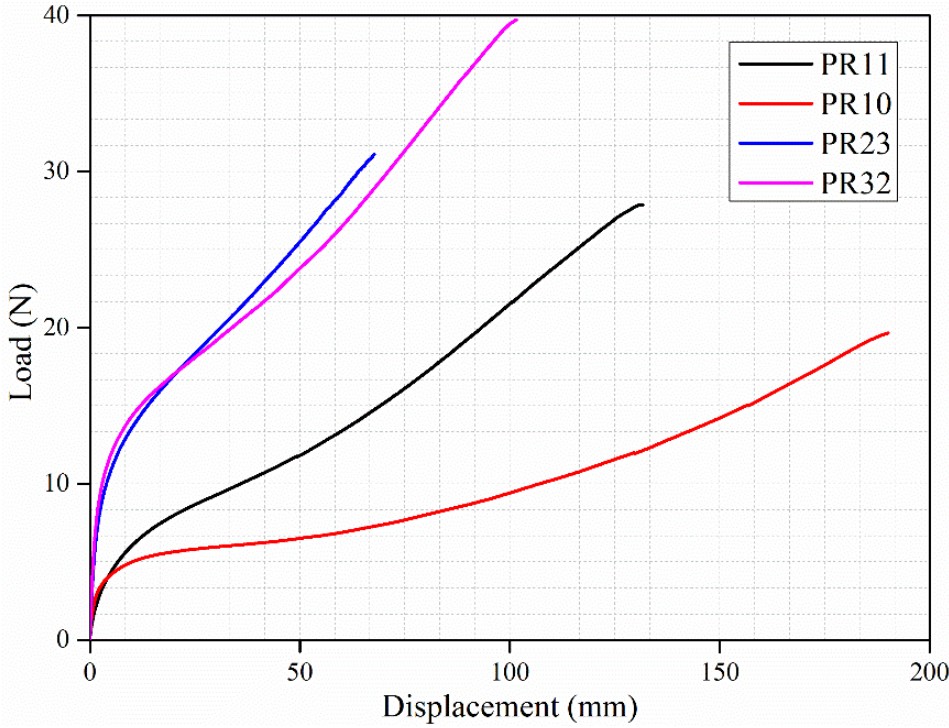

**Figure 5.** Tensile strength of composite modifier.

Figure 5 shows that the toughness of the modifier is very good when it is composed entirely of polyurethane. The displacement of PR10 film when completely broken is close to 200 mm. When 60% waste rubber powder is added to the composite modifier, its toughness decreases. The elongation of the PR23 when completely pulled is only about 60 mm. The curves of PR32 and PR11 are similar, but it seems that more work is needed to destroy PR32. To more accurately compare the differences between the different formulations, the energy required when the four composite modifiers are destroyed is summarized in Table 3. PR32 has the largest fracture energy, while PR23 has the smallest fracture energy. In other words, the influence of the content of polyurethane on the strength of the composite modifier is obvious. PR23 has the smallest percentage of polyurethane of the four formulations, so it has the lowest fracture energy. However, the cost of polyurethane is much higher than waste rubber powder, and it is hoped that more waste rubber powder is used in the composite modifier. After combining the fracture energy and economic profile considerations, PR11 was selected as the final composite modifier formulation.

**Table 3.** Tensile fracture energy of composite modifiers with different formulations.

| Samples | PR11 | PR10 | PR23 | PR32 |
|---|---|---|---|---|
| Fracture energy | 20,156 J/mm$^2$ | 19,262 J/mm$^2$ | 13,902 J/mm$^2$ | 25,122 J/mm$^2$ |

### 3.1.2. Microscopic Morphology

Figure 6 shows the microscopic morphology of waste rubber powder and the composite modifier. The surface of waste rubber powder is relatively rough, especially when the scanning multiple increases. This can be attributed to the fact that waste tires need to be broken several times when they are turned into rubber powder. However, the rough surface of waste rubber powder does not produce satisfactory compatibility with virgin asphalt. This can be explained by the fact that the compatibility of the modifier with the

asphalt is mainly related to factors such as polarity and solubility parameters, and the physical morphology is of secondary importance. The surface of the composite modifier is much smoother than that of waste rubber powder. This indicates that the waste rubber powder is tightly wrapped by polyurethane. Meanwhile, the raw material composition of polyurethane is very diverse, and its specific formula can be designed according to the solubility parameters of the matrix asphalt. Therefore, although the polyurethane film reduces the surface roughness of the composite modifier, it improves the compatibility between waste rubber powder and bitumen.

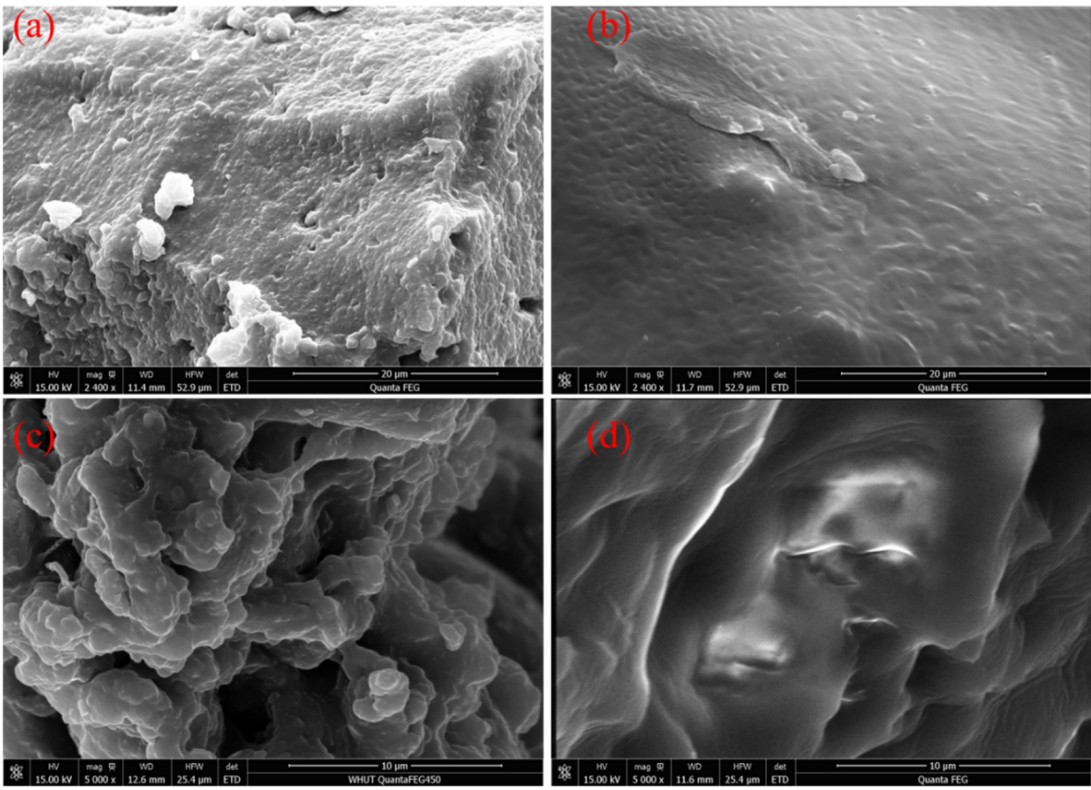

**Figure 6.** Microscopic morphology of waste rubber powder ((**a**) ×2400; (**c**) ×5000) and composite modifier ((**b**) ×2400; (**d**) ×5000).

*3.2. Physical Properties of Modified Asphalt Binders*

The effect of composite modifiers on the physical properties of asphalt binders is displayed in Figure 7. The softening point of modified bitumen increases, and its penetration decreases with the increase in the modifier dosage, as seen in Figure 7a. The 8 wt% composite modifiers can increase the softening point of virgin bitumen by 8.3 °C and reduce the penetration by 0.42 mm. Meanwhile, Figure 7b shows that the viscosity also increases, and the ductility declines sharply with the introduction of composite modifiers. Specifically, the viscosity of the asphalt binder increased by 0.07 Pa.s, 0.16 Pa.s, 0.27 Pa.s, and 0.38 Pa.s at composite modifier contents of 2 wt%, 4 wt%, 6 wt%, and 8 wt%, respectively, while the ductility of the bitumen decreased by 10%, 18.8%, 27.8%, and 36.9% at composite modifier contents of 2 wt%, 4 wt%, 6 wt%, and 8 wt%, respectively.

Both polyurethane and waste rubber powder are thermally stable polymers, which can still be filled in every corner of asphalt binders in the form of particles after high-temperature shearing. Therefore, when the two are combined to modify asphalt at the same time, the filling enhancement effect can still be better played, which leads to some improvements in the high-temperature characteristics of the modified bitumen, such as the increase in the softening point and viscosity. Similarly, the hardening of modified asphalt also has a slight negative impact on its flexibility, which is reflected in the performance of the reduction in ductility and penetration.

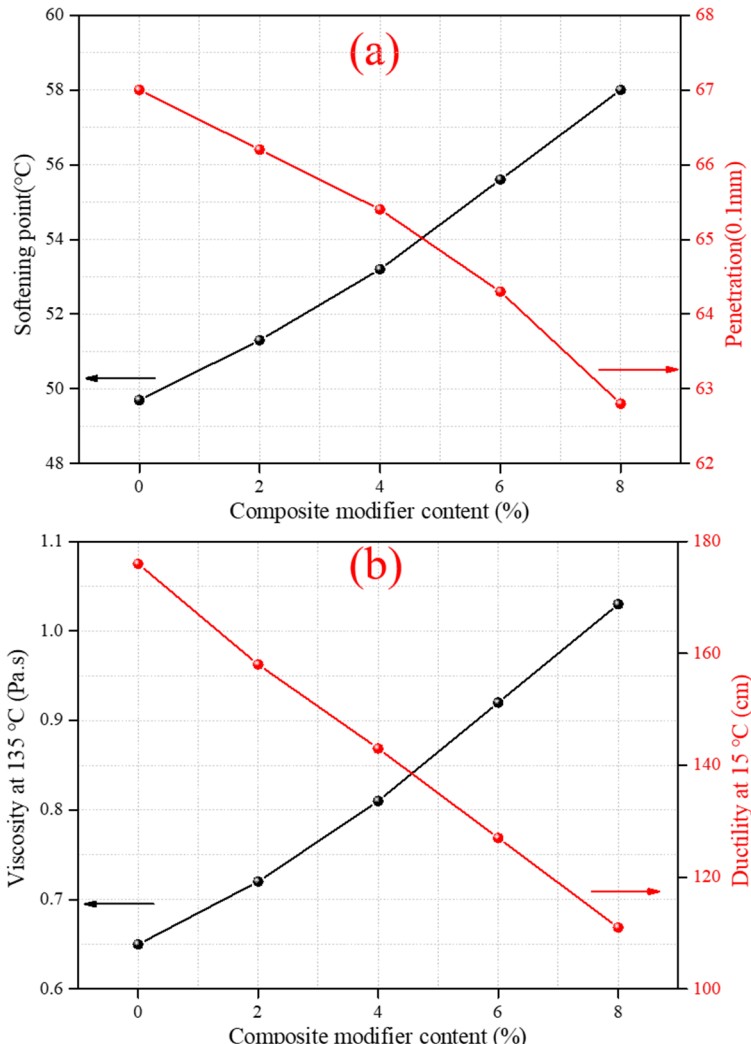

**Figure 7.** Physical properties of modified asphalt binders (**a**) softening point and penetration; (**b**) viscosity and ductility.

### 3.3. Engineering Performance of Modified Asphalt Mixtures

#### 3.3.1. Rutting Resistance

Permanent deformation is the most common damage suffered by asphalt pavements, especially in low-latitude countries. This is because as the temperature increases, the viscosity of bitumen decreases, and its ability to resist creep decreases, which can easily lead to the lateral flow of asphalt materials and cause rutting when subjected to external forces. Therefore, this section discusses the effect of polyurethane/waste rubber powder composite modifiers on the rutting resistance of asphalt mixtures. As displayed in Figure 8, the dynamic stability of the asphalt mixtures increases gradually with the increase in the composite modifier content. Specifically, the dynamic stability of the asphalt mixture increased by 12.7%, 22.1%, 32.2%, and 39.0% at composite modifier contents of 2 wt%, 4 wt%, 6 wt%, and 8 wt%, respectively. This means that the composite modifier can enhance asphalt mixtures' resistance to permanent deformation. It is generally accepted that the deformation resistance of asphalt mixtures is greatly affected by the asphalt binder. Asphalt binders with higher softening points do not soften easily, even in hot environments. In addition, bitumen with a higher viscosity is more likely to enclose aggregates. It was noted in the previous chapter that the filling effect of composite modifiers can improve the softening point and viscosity of asphalt binders. Therefore, the improvement in the rutting resistance of asphalt mixtures can also be attributed to the enhancement of the high-temperature performance of asphalt binders using composite modifiers.

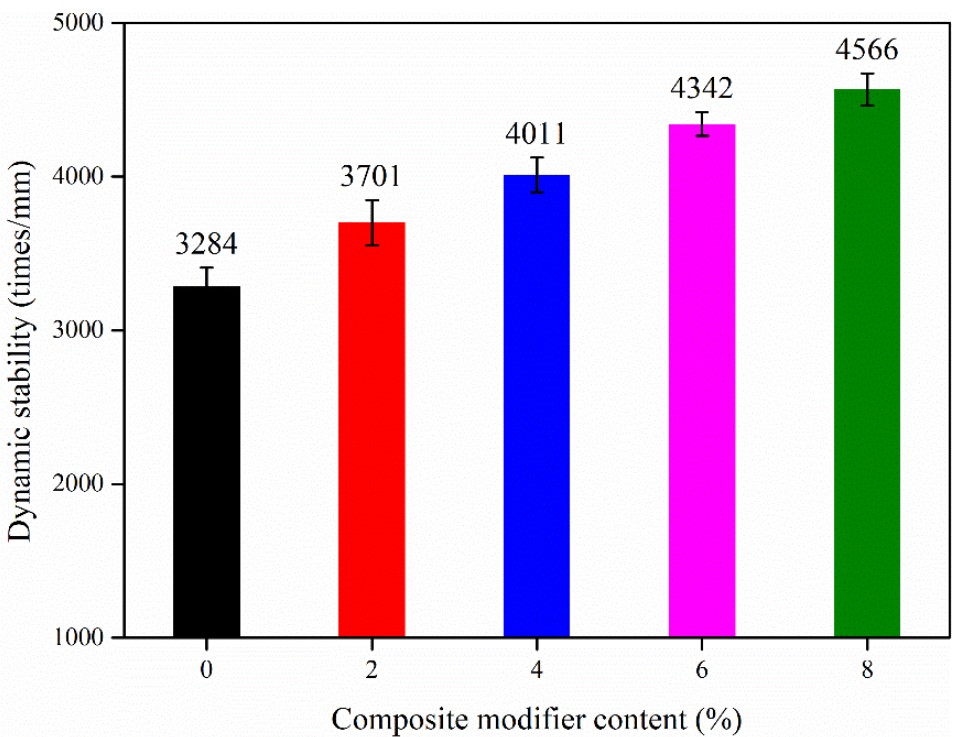

**Figure 8.** Dynamic stability of asphalt mixtures modified with composite modifier.

### 3.3.2. Cracking Resistance

Cracking is another one of the most common failures of asphalt pavement, particularly in severe cold areas. When the temperature drops abruptly, flexural strain is generated within the asphalt, which causes cracking in the pavement if it exceeds the ultimate flexural strain of the asphalt mixture. Therefore, the higher the ultimate flexural strain of the asphalt mixture, the lower the risk of cracking disease. In this section, the effect of composite modifiers on the low-temperature cracking resistance of asphalt mixtures is discussed.

Figure 9 shows that the flexural strain of the asphalt mixture first increases slightly and then decreases as the composite modifier content increases. Specifically, the flexural strain of the asphalt mixture increased by 1.6% and 2.7% at 2 wt% and 4 wt%, respectively, while the flexural strain of the asphalt mixture decreased by 1.8% and 4.9% at 6 wt% and 8 wt%, respectively. The result indicates that the low-temperature performance of asphalt mixtures can be enhanced if the composite modifier content is appropriate, while the anti-cracking properties of asphalt mixtures will be destroyed if the content of the composite modifier is too high. This can be understood as the bitumen gradually changes from soft to tough and then brittle with the increase in the modifier content. However, the flexural strain of the asphalt mixture containing 8 wt.% composite modifiers in this work is still 3367.43 $\mu\varepsilon$, which far exceeds the standard requirement of 2500 $\mu\varepsilon$. Overall, the specific dosage of the composite modifier should fully consider the low-temperature performance of asphalt mixtures.

### 3.3.3. Moisture Susceptibility

Moisture damage is currently one of the most common and destructive early failures of asphalt pavements. Moisture penetrates the interface between asphalt and aggregate, causing the asphalt film to peel away from the aggregate surface. The asphalt mixture gradually loses its internal cohesion, leading to other pavement diseases. Improving the resistance to moisture damage is critical to prolong the service life of asphalt pavements. Table 4 demonstrates the residual Marshall stability and the indirect tensile strength ratio of polyurethane/waste-rubber-powder-modified asphalt mixtures.

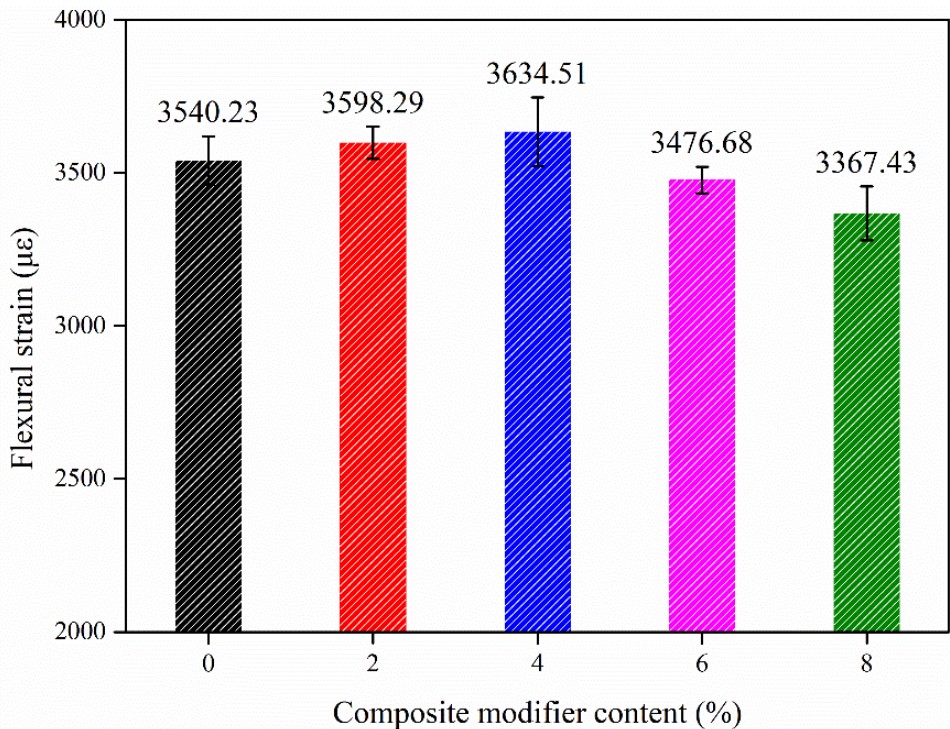

**Figure 9.** The tensile strain of asphalt mixtures modified with composite modifier.

**Table 4.** Residual Marshall stability and indirect tensile strength ratio of modified asphalt mixture.

| Modifier Content | $MS_0$ (Kn)/ Std. Dev | $MS_1$ (kN)/ Std. Dev | RMS (%) | $TS_0$ (MPa)/ Std. Dev | $TS_1$ (MPa)/ Std. Dev | ITSR (%) |
|---|---|---|---|---|---|---|
| 0 wt% | 14.25/2.61 | 12.73/2.14 | 88.0 | 0.981/0.054 | 0.837/0.037 | 85.3 |
| 2 wt% | 14.71/2.35 | 13.11/2.53 | 89.1 | 1.067/0.038 | 0.917/0.029 | 85.9 |
| 4 wt% | 17.95/2.87 | 16.16/2.77 | 90.0 | 1.090/0.042 | 0.943/0.036 | 86.5 |
| 6 wt% | 18.01/3.01 | 16.47/3.12 | 91.4 | 1.117/0.053 | 0.976/0.043 | 87.4 |
| 8 wt% | 18.92/2.93 | 17.35/3.02 | 91.7 | 1.132/0.049 | 1.015/0.051 | 89.7 |

As shown in Table 4, the original Marshall stability of the control specimen is 14.25 kN, and the Marshall stability of the asphalt mixture increases gradually with the increasing composite modifier content. Specifically, the Marshall stability can be increased by 32.8% at a composite modifier content of 8 wt%. This result indicates that the composite modifier is beneficial in improving the stability of the asphalt mixture. In addition, the *RMS* values of asphalt mixtures are also positively correlated with the composite modifier content, indicating that its resistance to moisture damage is also improved. This can be attributed to the composite modifier increasing the consistency of the virgin asphalt, which makes the bitumen become more tightly coated with the aggregate. Therefore, the erosion efficiency of moisture on asphalt mixtures is reduced.

On the other hand, Table 4 also demonstrates that the indirect tensile strength and *ITSR* value of the asphalt mixture are also increased. This means that the ability of the asphalt mixture to withstand the freeze–thaw cycle is enhanced by the composite modifier. This is because the moisture content in asphalt mixtures is an important factor affecting the occurrence of freeze–thaw cracking. The increased viscosity of the bitumen improves the adhesion of the asphalt to the aggregate, making it more difficult for moisture to penetrate the interior of the asphalt mixture. Therefore, the reduction in the water content of asphalt mixtures will reduce the volume change of water caused by the freeze–thaw cycle. In conclusion, the resistance of asphalt mixtures to moisture damage can be significantly improved via the introduction of composite modifiers.

## 4. Conclusions

The characteristics of polyurethane/waste rubber powder composite modifiers and their effects on the properties of asphalt mixtures were investigated in this work. The above analysis and discussion lead to the following conclusions:

The optimum mixing ratio of polyurethane to waste rubber powder is determined to be 1:1 based on the results of the tensile experiment and economic considerations. The tensile fracture energy of the composite modifier prepared using this formulation reaches 20,156 J/mm$^2$, which can cope with the mechanical mixing during the preparation of modified bitumen.

It is found that the surface of waste rubber powder is rough, while the surface of the composite modifier is smooth, which shows that the preparation method and formulation of the composite modifier in this paper can make the waste rubber powder become completely coated with polyurethane.

The results of the physical properties tests on the modified bitumen binders show that the composite modifier can significantly increase the softening point and viscosity of the bitumen while reducing the ductility and penetration of the bitumen, which is attributed to the filling effect of the composite modifier on the bitumen.

The composite modifiers can enhance the rutting and moisture damage resistance of asphalt mixtures by increasing the consistency of the asphalt binder, while the anti-cracking properties of asphalt mixtures will be destroyed if the content of the composite modifier is too high. It is suggested that a balance between the high- and low-temperature properties of asphalt mixtures should be fully considered when determining the content of composite modifiers in practical engineering.

**Author Contributions:** B.G., writing—original draft, methodology, and investigation; Y.Z., project administration, resources, supervision, and writing—review and editing; Z.Z., formal analysis and conceptualization. All authors have read and agreed to the published version of the manuscript.

**Funding:** This work was supported by the Hubei Science and Technology Innovation Talent and Service Project (International Science and Technology Cooperation) (2022EHB006) and the Key R&D Program of Guangxi Province (no. 2021AB26023).

**Data Availability Statement:** Data will be made available upon request.

**Conflicts of Interest:** The authors declare no conflict of interest.

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
