# Peer review of "Characteristics of Polyurethane/Waste Rubber Powder Composite Modifier and Its Effect on the Performance of Asphalt Mixture"

_sustainability, doi:10.3390/su151712703_

Round 1

Reviewer 1 Report

I found the topic of this research extra interesting, because it is way to become greener using old materials to new ones. Such researches are very important. I think that it is a good article to be published. Check line 91, for missing text.

Author Response

Dear Editor and Reviewers,

Thank you very much for your letter and the reviewers’ very helpful comments concerning our manuscript entitled “Characteristics of polyurethane/waste rubber powder composite modifier and its effect on the performance of asphalt mixture”. Those comments are all valuable and very helpful for improving our work. We have studied the comments carefully and have made modifications to the manuscript which we hope to meet with approval. Here, a revised manuscript with the corrected sections red marked was attached. And the details of the corrections are as follows. (The blue part is the reviewer’s comments, and the black part is our response)

I found the topic of this research extra interesting, because it is way to become greener using old materials to new ones. Such researches are very important. I think that it is a good article to be published. Check line 91, for missing text.

Response: Thank you very much for your constructive suggestions for our paper. We have corrected the error in line 91. We hope that our revisions will meet with your approval!

Sincerely,

Yuechao Zhao

E-mail: zhaoyc@whut.edu.cn

Reviewer 2 Report

Dear authors

The authors' efforts are appreciated, Bitumen additives are materials that are completely mixed with it under certain conditions in order to improve the properties or performance of the bitumen-containing additives or the asphalt mixture made with it. In general, bitumen containing additives is called modified bitumen. Polymers are the most important family of bitumen modifiers. Polymers increase the consistency of bitumen, especially at high temperatures when the bitumen flows so that the bitumen flows more slowly. These materials also increase the resistance of asphalt mixtures against fatigue and greatly increase the adhesion of bitumen to materials. Recovered rubber powder It has been widely used in bitumen modification. The use of rubber powder in asphalt can be considered more due to the rubber's elasticity and also its stiffness at ambient temperatures. The advantages of adding rubber powder to bitumen include: improving the fatigue resistance of asphalt mixtures, reducing She mentioned the noise caused by traffic, the reduction of environmental pollutants, and the improvement of the impact resistance of asphalt mixtures. Although the article has a proper laboratory research structure, some points need to be revised.

-In Figure 5, why didn't further deformation continue for pr23? To be explained., in other words, the fracture energy is low?

-Polyurethane is a polar chain, which can easily be mixed with other polymers and materials. How were these two polymers (with rubber powder) mixed? More mixing method or dry? add description.

- The combination of polymers has a positive effect on the behavior of bitumen, but how do you justify this effect on the rheological behavior of bitumen, the discussion on this matter should be explained more

-Why is the crack or fatigue behavior of bitumen chosen for -10 Celsius temperature? What was the standard test method selected?

-What is the standard reference for relationships 1 and 2? Please share.

- In Figure 6, what is the difference between Figure B and D and between Figure C and A? Be more specific under the title.

- In Figure 6, the horizontal axis should be written as a code for the percentage of each of the additives to the composite modifier.

-In Figure 7, how many percentages of positive or negative effects does the increase of "flexural strain" have on flexural? And are these percentages significant numbers in terms of statistical analysis? Perform statistical analysis of the data.

With respect

Author Response

Dear Editor and Reviewers,

Thank you very much for your letter and the reviewers’ very helpful comments concerning our manuscript entitled “Characteristics of polyurethane/waste rubber powder composite modifier and its effect on the performance of asphalt mixture”. Those comments are all valuable and very helpful for improving our work. We have studied the comments carefully and have made modifications to the manuscript which we hope to meet with approval. Here, a revised manuscript with the corrected sections red marked was attached. And the details of the corrections are as follows. (The blue part is the reviewer’s comments, and the black part is our response)

The authors' efforts are appreciated, Bitumen additives are materials that are completely mixed with it under certain conditions in order to improve the properties or performance of the bitumen-containing additives or the asphalt mixture made with it. In general, bitumen containing additives is called modified bitumen. Polymers are the most important family of bitumen modifiers. Polymers increase the consistency of bitumen, especially at high temperatures when the bitumen flows so that the bitumen flows more slowly. These materials also increase the resistance of asphalt mixtures against fatigue and greatly increase the adhesion of bitumen to materials. Recovered rubber powder It has been widely used in bitumen modification. The use of rubber powder in asphalt can be considered more due to the rubber's elasticity and also its stiffness at ambient temperatures. The advantages of adding rubber powder to bitumen include: improving the fatigue resistance of asphalt mixtures, reducing She mentioned the noise caused by traffic, the reduction of environmental pollutants, and the improvement of the impact resistance of asphalt mixtures. Although the article has a proper laboratory research structure, some points need to be revised.

-In Figure 5, why didn't further deformation continue for pr23? To be explained., in other words, the fracture energy is low?

Response: Thank you very much for your valuable comment. The strength of the composite modifier is largely dependent on the polyurethane content. PR23 has the smallest percentage of polyurethane of the four formulations, so it has the lowest fracture energy.

-Polyurethane is a polar chain, which can easily be mixed with other polymers and materials. How were these two polymers (with rubber powder) mixed? More mixing method or dry? add description.

Response: Thank you for your suggestion. Waterborne polyurethane is an emulsion in which polyurethane molecular chains are dispersed in water. Therefore, the waste rubber powder can be directly mixed in this emulsion system by mechanical agitation. With the evaporation of water, the emulsion will gradually solidify to form a composite material of polyurethane and waste rubber powder.

- The combination of polymers has a positive effect on the behavior of bitumen, but how do you justify this effect on the rheological behavior of bitumen, the discussion on this matter should be explained more

Response: Thanks for your kind suggestions. Rheological properties are indeed a point of concern for modified bitumen. However, this paper is a continuation of previous research (Y, Zhao. et al., 2022). Rheological parameters such as complex modulus, phase angle, rutting factor, and fatigue factor have already been carried out in the previous work. Therefore, we did not perform the same experiments in this manuscript.

[1] Y. Zhao, M. Chen, S. Wu, Q. Jiang, H. Xu, Z. Zhao, Y. Lv, Effects of waterborne polyurethane on storage stability, rheological properties, and VOCs emission of crumb rubber modified asphalt, Journal of Cleaner Production 340 (2022). http://dx.doi.org/10.1016/j.jclepro.2022.130682

-Why is the crack or fatigue behavior of bitumen chosen for -10 Celsius temperature? What was the standard test method selected?

Response: Thank you very much for your valuable comments. This experiment was carried out according to the Chinese standard JTG E20-2011 (T 0715). We have added this standard to the revised manuscript based on your comment.

-What is the standard reference for relationships 1 and 2? Please share.

Response: Thank you very much for the reminder. Relationships 1 and 2 refer to Chinese standard JTG E20-2011 (T 0709) and Chinese standard JTG E20-2011 (T 0729), respectively. An explanation has been added to the revised manuscript.

- In Figure 6, what is the difference between Figure B and D and between Figure C and A? Be more specific under the title.

Response: This is a good suggestion to improve the quality of our paper. The difference between them is mainly the scan multiple. We have made a more detailed note on the title.

- In Figure 6, the horizontal axis should be written as a code for the percentage of each of the additives to the composite modifier.

Response: Thank you very much for your valuable comments. We have already preferred the ratio of polyurethane to waste rubber powder to be 1:1 in the previous section. Therefore, all the composite modifiers in the following paper refer to PR11. In addition, Figure 6.

-In Figure 7, how many percentages of positive or negative effects does the increase of "flexural strain" have on flexural? And are these percentages significant numbers in terms of statistical analysis? Perform statistical analysis of the data.

With respect

Response: Thank you very much for your guidance. We have added statistical analyses in the revised draft. This advice is very helpful in improving the quality of our paper.

Finally, thank you very much for your constructive suggestions for our paper. We hope that our revisions will meet with your approval!

Sincerely,

Yuechao Zhao

E-mail: zhaoyc@whut.edu.cn

Reviewer 3 Report

The authors proposed an interesting topic “Characteristics of polyurethane/waste rubber powder composite modifier and its effect on the performance of asphalt mixture”. The conclusions of the study can provide a reference for the resource utilization of waste tire rubber powder and the green development of road engineering. However, some parts of the paper should be revised and the detailed comments are as follows:

1.     It is essential to evaluate some rheological properties of modified asphalt binders such as rutting factor and fatigue factor, etc.

2.     The preparation process of waterborne polyurethane is vague, so it is recommended to enrich the description of this part.

3.     It is suggested that the term “optimal mixing ratio of waterborne polyurethane and waste rubber powder" be replaced by "recommended mixing ratio of waterborne polyurethane and waste rubber powder”.

4.     Please unify the line spacing of tables.

5.     Why did the authors select the 2%, 4%, 6%, and 8% as the composite modifier dosage in the tests, please explain it.

Author Response

Dear Editor and Reviewers,

Thank you very much for your letter and the reviewers’ very helpful comments concerning our manuscript entitled “Characteristics of polyurethane/waste rubber powder composite modifier and its effect on the performance of asphalt mixture”. Those comments are all valuable and very helpful for improving our work. We have studied the comments carefully and have made modifications to the manuscript which we hope to meet with approval. Here, a revised manuscript with the corrected sections red marked was attached. And the details of the corrections are as follows. (The blue part is the reviewer’s comments, and the black part is our response)

The authors proposed an interesting topic “Characteristics of polyurethane/waste rubber powder composite modifier and its effect on the performance of asphalt mixture”. The conclusions of the study can provide a reference for the resource utilization of waste tire rubber powder and the green development of road engineering. However, some parts of the paper should be revised and the detailed comments are as follows:

  1. It is essential to evaluate some rheological properties of modified asphalt binders such as rutting factor and fatigue factor, etc.

Response: Thanks for your kind suggestions. Rheological properties are indeed a point of concern for modified bitumen. However, this paper is a continuation of previous research (Y, Zhao. et al., 2022). Rheological parameters such as complex modulus, phase angle, rutting factor, and fatigue factor have already been carried out in the previous work. Therefore, we did not perform the same experiments in this manuscript.

[1] Y. Zhao, M. Chen, S. Wu, Q. Jiang, H. Xu, Z. Zhao, Y. Lv, Effects of waterborne polyurethane on storage stability, rheological properties, and VOCs emission of crumb rubber modified asphalt, Journal of Cleaner Production 340 (2022). http://dx.doi.org/10.1016/j.jclepro.2022.130682

  1. The preparation process of waterborne polyurethane is vague, so it is recommended to enrich the description of this part.

Response: This is a very valuable suggestion. The synthesis method of waterborne polyurethane used in this work is described in detail in our published papers. We have made literature references where relevant. Therefore, the section is not overly described in the manuscript.

  1. It is suggested that the term “optimal mixing ratio of waterborne polyurethane and waste rubber powder" be replaced by "recommended mixing ratio of waterborne polyurethane and waste rubber powder”.

Response: This is a good suggestion to improve the quality of our paper. We have corrected this content in the revised manuscript. Thank you very much for your guidance.

  1. Please unify the line spacing of tables.

Response: We are sorry for this oversight. We have unified the line spacing of all tables in the revised manuscript.

  1. Why did the authors select the 2%, 4%, 6%, and 8% as the composite modifier dosage in the tests, please explain it.

Response: Thanks for your valuable comment. Previous work has found that agglomeration occurs when the dosage of composite modifier exceeds 8%, which is detrimental to the low-temperature performance and fatigue performance of asphalt (Y, Zhao. et al., 2022). Therefore, the dosage of composite modifiers in this work did not exceed 8%. Additionally, the dosage gradient of 2% is set to compare the difference in the effect of composite modifier dosage on the performance of asphalt mixtures. Therefore, we selected 2%, 4%, 6%, and 8% as the composite modifier dosage in the tests.

Finally, thank you very much for your constructive suggestions for our paper. We hope that our revisions will meet with your approval!

Sincerely,

Yuechao Zhao

E-mail: zhaoyc@whut.edu.cn